# Prediction of Esterification and Antioxidant Properties of Food-Derived Fatty Acids and Ascorbic Acid Based on Machine Learning: A Review

**DOI:** 10.3390/foods14244255

**Published:** 2025-12-10

**Authors:** Xinyu Wang, Jianyi Wang, Xiaoyu Zhang, Tiantong Lan, Jingsheng Liu, Hao Zhang

**Affiliations:** Corn Processing Laboratory of China Agriculture Research System, College of Food Science and Engineering, Jilin Agricultural University, Changchun 130118, China; wangxinyu173@163.com (X.W.); wjy1040340736@163.com (J.W.); zhangxiaoyu0875@163.com (X.Z.); liujingsheng@jlau.edu.cn (J.L.)

**Keywords:** model prediction, random forests, free radical scavenging, ascorbyl ester

## Abstract

This study is dedicated to summarizing and performing an in-depth analysis of the antioxidant properties of ascorbic acid fatty acid esters. The esterification reaction mechanism of ascorbic acid with palmitic acid, lauric acid, and oleic acid in food systems was elaborated in detail, and its antioxidant mechanism was discussed in depth. The free radical scavenging mechanism and oxidative inhibition effect of two mainstream determination methods, DPPH and ABTS, were analyzed. Esterification, as a core organic synthesis reaction, is widely used in the production of food antioxidants, pharmaceutical ingredients, chemical polymers, and cosmetic oil-based matrices. At the same time, in view of the wide application of machine learning as a multidisciplinary core technology, this paper selects free radical scavenging rate and esterification yield as characteristic parameters and normalizes the offspring into random forest model training to achieve accurate prediction of antioxidant performance. Finally, in the future, it is necessary to expand the data set, optimize the model structure, explore multi-model fusion to improve the prediction effect, and promote the application of machine learning in the screening design of new antioxidants and the optimization of green synthesis processes to promote the intelligent and sustainable development of food antioxidant research.

## 1. Introduction

Nowadays, chronic diseases such as neurodegenerative diseases and metabolic syndrome occur frequently. As a key link in disease prevention and auxiliary conditioning, dietary components have attracted much attention for their functions. In meat products, fruit and vegetable products, and oil and fat foods, inhibiting lipid peroxidation and polyphenol oxidase activity is particularly important for effectively delaying food browning and rancidity, as well as prolonging shelf life. Ascorbic acid (VC) as a natural antioxidant can replace the potential risk of synthetic antioxidants. It is an essential water-soluble vitamin for humans, synthesized from glucose through a series of enzymatic reactions [1]. VC can be extracted from plants, fruits, vegetables, and algae, and its content of VC in natural foods (10~100 mg/100 g) is usually high, and the body itself cannot synthesize this nutrient, so it is necessary to meet the body’s needs through dietary intake [2]. VC can directly react with reactive oxygen species (ROS) and free radicals (superoxide anion, hydroxyl radical, etc.). It provides electrons to achieve an antioxidant effect. While it is oxidized, it allows ROS to obtain electrons, thereby improving the stability of ROS and reducing its reactivity. Finally, it avoids excessive accumulation of ROS, causing damage to cells [3]. VC can also maintain intracellular redox balance by reducing other oxidizing substances, such as dehydroascorbic acid (DHA) [4]. VC protects the integrity of cell membranes, proteins, and nucleic acids by scavenging free radicals, as well as reducing lipid peroxidation, protein oxidation, and DNA damage [2].

Although free VC has high antioxidant activity, its chemical properties are unstable and easy to oxidize and inactivate. Fatty acids (FA) themselves have no significant antioxidant activity. However, the ester products formed by the esterification of FA with VC can achieve synergistic enhancement of antioxidant efficiency through a dual mechanism [5]. The hydrophobic interaction of fatty acid chains can wrap the active hydroxyl groups (-OH) in VC molecules, reduce the contact probability of active sites with oxygen and free radicals, and thus reduce the oxidation inactivation rate of VC. FA can be divided into saturated fatty acids (SFA) and unsaturated fatty acids (UFA). There is no double bond in SFA, and there are single bonds between carbon atoms. Common SFAs are palmitic acid, lauric acid, and stearic acid, etc. UFAs are divided into monounsaturated fatty acids (MUFA) and polyunsaturated fatty acids (PUFA) according to the double bond. MUFAs contain one double bond, and PUFAs contain two or more double bonds. Common UFAs are oleic acid, linoleic acid, α-linolenic acid, γ-linolenic acid, etc. [6]. FAs are the key energy storage molecules in the human body and perform a variety of important functions throughout the life cycle. As the core component of lipids, FAs not only participate in energy metabolism and maintain the stability of cell membranes but also play a regulatory role in various physiological processes of cells [7].

UFAs help to reduce the risk of cardiovascular disease, lower triglyceride levels, improve endothelial function, and reduce thrombosis, while exerting a positive impact on inflammation and lipid metabolism [8]. Additionally, it also has a certain effect on improving type 2 diabetes, as they can enhance glucose utilization and absorption, as well as insulin secretion and function [9]. However, excessive intake of SFAs may cause inflammation, leading to chronic diseases such as obesity and type 2 diabetes. In addition, SFA increase the risk of cardiovascular disease by increasing the level of low-density lipoprotein cholesterol in the blood [10]. However, since SFAs do not contain double bonds, compared with UFAs containing one or more double bonds, their chemical structure is more stable and less prone to oxidative deterioration.

To prevent the deterioration of fats, proteins, and cellular DNA, which can lead to undesirable odors and tastes in FA, natural or synthetic antioxidants are usually used to scavenge free radicals and inhibit the activity of oxidase [11]. Both VC and FA are natural substances in organisms, and their esterification products exhibit high safety. They can be directly applied to food, cosmetics, medicine, and other fields related to the human body. Due to VC’s poor fat solubility, it is difficult for it to exert its antioxidant effect in oil-based systems or the cellular lipid phase. FAs have typical fat solubility. After introducing fatty acid chains into VC molecules by esterification reaction, their fat solubility can be significantly improved so that the products can be dispersed in both the water phase and the fat phase system to achieve ‘biphasic antioxidant coverage’ [12]. Ascorbate captures free radicals and reacts with free radicals to form relatively stable compounds, preventing further oxidation reactions, thereby achieving antioxidant effects.

In recent years, machine learning (ML) has been widely used in the food industry and has many advantages in identifying commonalities and distinguishing sample feature differences [13]. In order to realize the rapid and accurate evaluation of the antioxidant activity of ascorbyl fatty acid esters with different molecular structures or parameters, an ML-based antioxidant performance prediction model can be constructed to provide quantitative theoretical support for the optimization of the directional synthesis process of highly active ascorbyl fatty acid esters. As a branch of artificial intelligence, ML can train data models, infer patterns, and optimize parameters by leveraging the similarity of existing examples, thereby predicting new data, compared with traditional algorithms that rely on explicitly programmed instructions or rules [14]. ML includes supervised learning, unsupervised learning, and deep learning. Supervised learning is suitable for labeled data, while unsupervised learning is suitable for unlabeled data. Deep learning is suitable for processing data with complex, large-scale, and nonlinear relationships [15]. By comparing the advantages of supervised learning and unsupervised learning, Zhao et al. [16] found that supervised learning can save resources and time, possesses strong learning capabilities, and can effectively utilize multiple data models to improve prediction accuracy.

This review mainly discusses the process of esterification of VC with different FAs, as well as the antioxidant properties and mechanisms of the generated VC esters. Through the cross-application of ML and food science, the antioxidant properties of ascorbate are accurately predicted, which provides new ideas for research and application in related fields. Figure 1 below condenses the core of the review by means of a visual workflow.

## 2. The Process of Esterification of Ascorbic Acid and Fatty Acid

As the basic constituent unit of fat, FA is an essential nutrient for the human body. It exhibits great diversity. According to the length of the carbon chain, FA can be divided into short-chain FA (carbon chain contains 4–6 carbon atoms, such as acetic acid and propionic acid), medium-chain FA (carbon chain contains 8–12 carbon atoms, such as octanoic acid and decanoic acid), and long-chain FA (carbon chain contains 14 or more carbon atoms, such as palmitic acid and stearic acid). Among many FAs, OA, PA, and LA are selected as the key introduction objects, mainly based on the representativeness and importance of these three in food science, nutrition and health, and industrial applications. From the perspective of structure and classification, they represent the typical characteristics of MUFA, long-chain SFA, and PUFA, respectively, covering the core categories of FA. Through their research, the differences in esterification properties of FA with different saturations and different carbon chain lengths can be clearly demonstrated. In terms of practical application value, PA is widely found in palm oil and animal fat. It is an important component of oil commonly used in the food industry and has an important impact on the taste and shelf life of food [17]; as the main functional component of olive oil and tea seed oil, OA has good nutritional characteristics, and related research has attracted much attention in the field of nutrition and health. LA is widely used in food additives, cosmetics, and the pharmaceutical industries and is widely found in animal fats and vegetable oils. The main sources of these three FAs and the esterification reaction process of VC with three FAs are shown in Figure 2. In the figure, the oxygen atom marked by the red box in the VC molecule is the site that binds to FA, while the part marked by the blue box corresponds to the functional group that is eliminated as a water molecule during the reaction. In the esterification reaction, the hydroxyl group (-OH) of VC reacts with the carboxyl group (-COOH) of FA to form an ester group (-COO-) and a water molecule (H_2_O). This reaction reflects the chemical combination between VC and FA to generate esters with antioxidant properties, which plays an important role in the fields of food, cosmetics, and medicine.

Esters occupy a key position in the field of food additives, and their applications widely cover multiple sub-scenes in the food industry. At present, food additive manufacturers are gradually moving away from the traditional chemical synthesis path and towards greener and more efficient alternative processes. Among them, biocatalytic synthesis technology based on immobilized enzymes stands out for its unique advantages and has become one of the research hotspots and a core development direction in this field. Numerous research papers included in the Web of Science (WOS) database further confirm that the development of more sustainable new synthesis processes of ester is of great theoretical and practical significance for promoting the green transformation of the food industry and meeting the high-quality development needs of the industry [18]. A variety of fruit flavors in the food industry rely on esterification preparation, and the production of esters can be optimized by regulating fermentation conditions to enhance flavor. And it can effectively improve the food processing characteristics and quality of food. The medium-chain triglyceride produced by esterification is a high-quality nutrient carrier and can be supplied quickly. The enzymatic esterification technology in oil processing can avoid the formation of trans fat, thereby taking into account both the taste and health of food. In the field of food preservation and packaging, ester compounds synthesized by esterification can be used as preservatives to inhibit microbial reproduction. At the same time, it also promotes the green development of the food industry. Compared with the high energy consumption and high pollution of traditional chemical synthetic esters, enzyme-catalyzed esterification technology is more environmentally friendly and provides support for the large-scale production of functional esters.

### 2.1. Palmitic Acid

Palmitic acid (PA), whose chemical formula is C_16_H_32_O_2_, is a kind of saturated fatty acid, that is very present in nature. It is mostly found in animals and plants, algae, fungi, yeast, and bacteria, and the content in vegetable oil is 10–45% [19]. PA is also one of the SFAs widely present in the human diet. It is not only an important energy source for lipid metabolism but also a key component of human lipid composition [20]. By maintaining the stability and fluidity of cell membranes, PA plays an essential role in ensuring the normal physiological functions of cells [21]. Ascorbyl palmitate synthesized by VC and PA can be used as a compatibilizer, which has amphoteric properties with a hydrophilic head and hydrophobic fatty acid tail [22]. At the same time, it can also delay lipid oxidation by scavenging free radicals as an antioxidant [23].

Compared with VC, ascorbyl palmitate has better thermal stability and can enhance the antioxidant properties of vegetable oils during heating [24]. Ascorbyl palmitate (AP), whose chemical formula is C_22_H_38_O_7_, is soluble in ethanol, a fat-soluble derivative of VC. AP is a compound formed by esterification of VC and PA. It is a white or yellow-white powder with a citrus aroma and is an amphiphilic molecule. It contains a hydrophobic side (alkyl chain) and a hydrophilic polar side (VC) [25]. AP has similar antioxidant properties to VC, which reduces oxidative damage to cells by providing electrons to neutralize free radicals [25]; therefore, it is widely used as an antioxidant in the food, pharmaceutical, and other industries [26].

The synthesis of AP is divided into two processes: chemical synthesis and enzymatic synthesis, the former being the most common method. The technical method of chemical synthesis of ascorbyl ester is relatively more mature and low cost, which is suitable for large-scale production. However, due to its harsh reaction conditions, it needs strong acid, alkali, and other reagents for treatment [27]. It may cause corrosion of the instrument and produce certain pollutants such as waste gas and wastewater, which will cause certain pressure on the environment. Yadav et al. [28] the enzymatic synthesis of AP is divided into esterification reactions in batch mode and continuous mode. Batch-mode esterification is more flexible in cases involving small-scale production, laboratory research, frequent product type changes, or adjustments to reaction conditions. Continuous mode esterification is more suitable for large-scale production, higher requirements for product quality and production efficiency, and sufficient capital investment in equipment and process development experiments. The purity of the product of the enzymatic reaction is high, and there is basically no by-product formation. Compared with chemical synthesis, the reaction process is more environmentally friendly and meets the requirements of sustainable development [29].

Chemical synthesis usually requires harsh synthesis conditions, such as strong acid and alkali, high temperature, and high pressure, etc., which may lead to a variety of by-products and reduce the efficiency of the synthesis reaction. Compared with chemical synthesis, the enzymatic reaction has high specificity. Lipases are classified based on their regiospecificity and the positions of ester bonds they target in triglycerides. Sn-1,3-specific lipase hydrolyzes ester bonds at sn-1 and sn-3 positions, sn-2-specific lipase hydrolyzes ester bonds at sn-2 positions, and non-specific lipase hydrolyzes all positions of triglyceride randomly [30]. The enzymatic reaction is usually carried out under mild conditions. The optimum pH is generally 4–9, and the optimum temperature is 25–70 °C, which reduces the formation of intermediate products and by-products, thereby increasing the reaction yield [31].

### 2.2. Lauric Acid

Lauric acid (LA), also known as dodecanoic acid [32], whose chemical formula is C_12_H_24_O_2_, is a medium-chain saturated fatty acid, mainly derived from natural vegetable oil, and is the main component of tropical oils such as coconut oil and palm oil [33]. Due to its unique chemical structure, LA has significant metabolic advantages in the human body. As a medium-chain fatty acid, it has a small molecular weight and can be quickly absorbed and metabolized by the human body. It is mainly converted into energy in the body, rather than easily resynthesizing triacylglycerol and storing it as fat, like long-chain FA [34]. LA has significant antibacterial properties, and its derivatives inhibit the growth of microorganisms and regulate the non-fatty acid part of the molecule, their own molecules [35]. In addition, LA also has the effect of regulating blood lipids, which can increase the level of high-density lipoprotein, thus helping to reduce the risk of atherosclerosis [34]. At the same time, LA also has a certain reductive property, which can effectively remove free radicals in the body, thereby reducing the oxidation reaction. The functional properties of LA, as well as the structural advantages of medium-chain FA, provide an ideal site for chemical modification. Through the esterification reaction with VC, the hydrophilic skeleton of VC can be introduced into the alkyl chain of LA to construct the lipophilic and hydrophilic amphiphilic molecule of ascorbyl laurate, which lays a structural foundation for expanding its functional application.

Ascorbyl laurate (AL), whose chemical formula is C_18_H_30_O_7_, is slightly soluble in water, usually in the form of white crystalline powder. VC laurate is an amphiphilic substance produced by the esterification of VC and LA. It has various properties, such as antioxidant and antibacterial properties [36]. This amphiphilic property allows it to act as an emulsifier to enhance the surface activity of the solution and the high foaming stability, thereby forming a stable oil-in-water (O/W) emulsion [37]. At the same time, AL can also be used as an antioxidant to delay lipid oxidation in oil-in-water emulsions. It can concentrate the antioxidant molecules at the oil-water interface, which is the area where the oxidation is more concentrated in the emulsion, so it can more effectively block lipid oxidation [38]. In addition, experiments have confirmed that AL has a good antibacterial and bactericidal effect on Gram-positive foodborne bacteria [38]. It is mainly used as an antioxidant, emulsifier, and antibacterial agent in food, which can prolong the shelf life, maintain food stability, and inhibit the growth of pathogens. And EL also has a certain vitamin C activity, which can be used to strengthen nutrition, so it is widely used in many foods. AL is usually produced by an enzyme-catalyzed reaction. The enzyme catalyst has high specificity and selectivity, basically no by-products are generated, and the purity of the product is improved. Enzyme-catalyzed synthesis has good operational stability, the reaction conditions are easy to control and optimize, and it is suitable for a variety of reaction systems. Its mild reaction conditions reduce the safety risks and energy consumption caused by harsh conditions. Therefore, enzyme-catalyzed synthesis is not only environmentally friendly but also improves product quality and production efficiency [39].

Hyunjong Yu et al. [40] in the experiment the yield and stability of the synthesis of AL in an organic solvent single-phase system (OS-MPS), a solid–liquid two-phase system (SL-BPS), and a gas–solid multiphase system (GSL-MPS) were compared, and it was found that OS-MPS had the highest reaction rate compared with other reaction systems, but its yield was the lowest among all reaction systems. There was no significant difference in yield between SL-BPS and OS-MPS at the same reaction time, while the yield of GSL-MPS was much higher than that of SL-BPS at the same reaction time. At the same time, the stability of the immobilized lipase after reaction in OS-MPS was extremely poor, and the enzyme activity after treatment was reduced to one-tenth of the original, while the lipase activity in GSL-MPS gradually increased during the treatment process. Therefore, it can be found that GSL-MPS has significant advantages in solvent-free synthesis, which can improve the reaction efficiency and product yield and enhance the stability and utilization of the enzyme.

### 2.3. Oleic Acid

Oleic acid (OA), whose chemical formula is C_18_H_34_O_2_, is a colorless, oily liquid, almost insoluble in water. OA accounts for 40–83% of peanut oil, rapeseed oil, olive oil, and other vegetable oils [41]; it is the main component of these vegetable oils. As an important nutrient component, it is a monounsaturated fatty acid required by the human body with many functional properties, such as antioxidation, antibacterial, cholesterol reduction, and cardiovascular protection [42]. Because OA can provide acyl, it has strong hydrophobicity and good antioxidant and antibacterial properties [43]. The long-chain hydrophobic skeleton of OA provides a structural template for esterification with VC, and the two are esterified to form ascorbyl oleate with both amphiphilic and antioxidant and antibacterial functions.

Ascorbyl oleate (AO), whose chemical formula is C_26_H_40_O_7_, is usually a light yellow oily substance, which is a multifunctional compound. It is widely used in the food industry as an antioxidant and nutritional supplement, and its antioxidant and anti-aging effects enable it to be used in cosmetics development. It can also improve drug delivery efficiency in pharmaceutical research and development and preparation of nanocarriers and has important application value. The DPPH free radical scavenging ability of AO is higher than that of ABTS free radical scavenging ability, but its antioxidant capacity is weaker than that of VC. At the same time, it can inhibit the growth of Gram-positive bacteria and Gram-negative bacteria, which also reflects its antibacterial ability as a food preservative [44]. AO is synthesized by VC and OA catalyzed by immobilized lipase. Ha-Eun Ji et al. [45] effects of two synthesis methods catalyzed by lipase on the purity and yield of OA were compared by experiments. The two methods were lipase-catalyzed esterification synthesis and lipase-catalyzed transesterification synthesis.

Lipase-catalyzed esterification is a method to synthesize AO via a lipase-catalyzed esterification reaction. Using Novozym 435 (Novozymes, Bagsvaerd, Denmark) as a catalyst, the yield of the synthesized product is as high as 78.2% [44]. Compared with the traditional solvent-free reaction system, the GSL-MPS reaction system uses OA as the liquid-phase reaction medium and uses lipase to catalyze the esterification reaction of VC and OA. Due to the enhanced dispersion of insoluble VC in the reaction medium, the esterification reaction product will increase significantly. In addition, it can effectively overcome the mass transfer limit between solid-phase VC and liquid-phase FA in the gas phase [46].

Lipase-catalyzed transesterification synthesis is the synthesis of AO by transesterification catalyzed by lipase Novozym 435. The reaction is usually carried out in organic solvents (such as acetone), using lipase as a catalyst. Studies have shown that due to the optimization of reaction conditions, reaction temperature, substrate molar ratio, catalyst dosage, and other factors have a significant impact on the reaction yield [45]. The transesterification reaction has become a green, efficient, and sustainable chemical reaction method due to its mild reaction conditions, high selectivity, and environmental friendliness [47].

Esterification is usually used to synthesize esters from carboxylic acids and alcohols, while transesterification is used to synthesize new esters from existing esters and alcohols. Ha-Eun Ji et al. [45] developed an economical and eco-friendly synthesis approach with acetone as the solvent, and this method can remarkably enhance the yield of AO. At the same time, impurities in the product were effectively removed, and the purity was improved by solvent extraction, washing, recrystallization, and liquid–liquid extraction. Studies have shown that the synthesis efficiency of direct esterification of VC and OA is low. Lipase-catalyzed transesterification of AP with OA can significantly increase the synthesis rate, thereby increasing the yield of the target product.

## 3. Antioxidant Properties and Mechanism

Issa Javidipour et al. [48] studied the effects of AP on the peroxide value of refined cottonseed oil and virgin olive oil under specific conditions (reflecting the peroxide content produced in the early stage of oil oxidation, which is the core index to evaluate the degree of oil oxidation) and malondialdehyde content (reflecting the content of aldehydes produced in the later stage of oil oxidation, which can characterize the accumulation of secondary products of oil oxidation). They showed that compared with refined cottonseed oil and virgin olive oil, the two oils after adding AP had lower peroxide values than the control oils, and the peroxide value was an indicator of the degree of oil oxidation. The lower the peroxide value, the stronger the antioxidant performance. At the same time, compared with refined cottonseed oil and virgin olive oil, the two oils after adding AP were lower than their individual malondialdehyde content, and the level of malondialdehyde content represents the level of oxidative stress. The low content indicates its strong antioxidant capacity, which can effectively scavenge free radicals and prevent the occurrence of lipid peroxidation. This is because the antioxidant activity of ascorbate is closely related to its enediol structure. The enediol structure is relatively prone to oxidation and can provide hydrogen atoms or electrons to neutralize free radicals, thereby inhibiting the oxidation reaction [49]. Specifically, the enediol structure (hydroxyl groups of C-2 and C-3) in VC can improve the antioxidant properties of VC ester after esterification with liquid-phase FA in a fat-soluble environment. Marija Stojanović et al. [50] used the DPPH free radical scavenging method to determine the antioxidant activity of the synthesized and purified ester products, L-ascorbic acid, and commercial-grade AP. The absorbance value of the reaction system was determined by spectrophotometer, and the free radical scavenging rate of each sample was calculated according to the absorbance change. The IC_50_ value (half inhibitory concentration) was used as the core evaluation index of antioxidant activity, which was defined as the minimum concentration of antioxidants required to reduce the initial concentration of DPPH free radical by 50%. The smaller the IC_50_ value, the stronger the antioxidant activity of the sample. The ABTS radical cation decolorization assay was to generate ABTS^•+^ by chemical oxidation reaction of potassium persulfate, and the absorbance at 734 nm was measured by microplate reader; its oxidation resistance is shown by calculations [51]. DPPH and ABTS can indirectly reflect the ability of antioxidants to scavenge natural ROS by electron or hydrogen transfer reaction with antioxidants. Because of their high homology with ROS in structure and reaction mechanism, they have become the core in vitro tools for evaluating the antioxidant properties of natural antioxidants or composite carrier materials in the fields of food science, nutrition, and materials science and provide important reference for subsequent in vivo ROS scavenging experiments in vivo.

### 3.1. Free Radical Scavenging Method

According to different free radicals, different methods were used to determine the final oxidation products and analyze the antioxidants. The most common free radical removal methods were 2,2’-azino-bis (3-ethylbenzothiazoline-6-sulfonic acid) (ABTS), 2,2-diphenyl-1-trinitrophenylhydrazine (DPPH), and N,N-Dimethyl-p-phenylenediamine (DMPD) spectrophotometry [52].DPPH^•^ + AH→DPPH_2_ + A^•^ABTS^•+^ + AH→ABTS^+^ + A^•^DMPD^•+^ + AH→DMPD^+^ + A^•^

The two most commonly used methods for the determination of antioxidant capacity are DPPH and ABTS [53]. Due to the different mechanisms of action of the two methods on free radicals, a single determination method may not fully reflect the antioxidant capacity of the sample. Therefore, the combination of two or more determination techniques to determine the ‘total antioxidant capacity’ can more comprehensively and objectively present the antioxidant properties of the sample [44].

The DPPH free radical scavenging method and the ABTS free radical scavenging method are used as commonly used antioxidant activity determination methods. Compared with the ABTS free radical scavenging method, the DPPH free radical scavenging method is simpler and cheaper, and the reaction conditions are mild. The free radical scavenging effect can be expressed by color change [54]. It is suitable for preliminary screening and comparing the antioxidant activity of different samples, especially for the detection of water-soluble and fat-soluble antioxidants. Due to its strong specificity [55], it is also suitable for the rapid screening of fat-soluble antioxidants and can accurately reflect the scavenging ability of antioxidants on singlet free radicals. However, because the DPPH method is sensitive to the interference of colored substances, it may affect the accuracy of the results. Therefore, in practical applications, it is usually necessary to combine the two methods for comprehensive evaluation to obtain more comprehensive and accurate results.

### 3.2. DPPH Free Radical Scavenging Mechanism

A free radical is a special chemical entity, and its core feature is that it has an unpaired electron. The latter has unique spin quantum mechanical properties, which endow free radicals with high reactivity and special chemical behavior, which makes free radicals prone to oxidation reactions [56]. Since free radicals have harmful effects on food, free radical activity should be eliminated to ensure food quality and delay their oxidative deterioration. Because the chemical structure of DPPH contains a nitrogen atom, the unpaired electrons on the nitrogen atom give DPPH unique free radical properties, so it has become a stable free radical molecule widely used in the study of antioxidant properties. DPPH has a strong absorption at 517 nm. After being reduced by antioxidants, it fades from deep purple to light yellow and finally becomes colorless (Figure 3). The ability of the sample to scavenge free radicals can be quickly and sensitively evaluated by measuring the degree of absorbance decline. Many assays are used to evaluate the antioxidant activity of herbal extracts or phenolic compounds [52].

The antioxidant activity of ascorbyl fatty acid esters is mainly attributed to the free hydroxyl groups at the C-2 and C-3 positions of the ascorbyl moiety. These hydroxyl groups act as reducing agents to neutralize free radicals by transferring hydrogen atoms and electrons or as deoxidizers to react with oxygen. Since the esterification reaction mainly occurs at the C-6 position, these hydroxyl groups remain active after esterification with liquid-phase FA [57]. Ascorbyl fatty acid esters are fat-soluble compounds and have reducibility, which can transfer electrons to unpaired electrons on the nitrogen atoms of DPPH free radicals so that their single electrons are paired and converted into stable DPPH_2_ molecules, as shown in Figure 2. Ascorbate can also scavenge DPPH free radicals by transferring protons. DPPH exists in solution as a dynamic equilibrium between the free ground state (DPPH·) and the positive ion state (DPPH^+^). VC ester can transfer protons to DPPH, making it into DPPH_2_, thereby reducing the concentration of DPPH in the solution and achieving free radical scavenging. VC ester can react with DPPH^+^ via a redox reaction. DPPH^+^ has a highly conjugated nitrogen–nitrogen double bond structure. VC ester can reduce the nitrogen–nitrogen double bond of DPPH^+^ and decompose it into trinitroaniline and other products to achieve the effect of scavenging DPPH free radicals [58].

As shown in Figure 4, the conversion processes between different ROS forms a complex redox network, maintaining the balance between intracellular oxidation and antioxidation. When this balance is broken, it will lead to oxidative stress. The natural oxidation of food components and the oxidation during processing will lead to oxidative stress in food, which will lead to poor flavor, color, and texture changes; loss of nutrients; reduced bioavailability; and adverse effects on human health.

Markus et al. [59] studied five reaction mechanisms of DPPH, including hydrogen atom transfer (HAT), proton transfer (PT), single electron transfer (SET), sequential proton loss electron transfer (SPLET), and single electron transfer-proton transfer (SET-PT). In the HAT mechanism, a complete hydrogen atom (including its electrons) are transferred to the DPPH free radical and reduced to DPPH-H, while the antioxidant molecule itself is converted into a free radical. In order to prevent further reaction, the free radical needs to be stabilized by means of delocalization or steric hindrance. The mechanism of PT is similar to that of HAT. Because only protons are transferred, protonated DPPH-H and antioxidant molecular anions are generated. The reaction process is slow and is usually affected by pH value. In the SET mechanism, electrons is transferred from the antioxidant to the DPPH free radical so that the DPPH free radical is reduced to DPPH, while the antioxidant itself is oxidized. This mechanism is a rapid reaction mechanism, usually manifested as a rapid decrease in the initial absorbance. The reaction is strongly dependent on the nature of the solvent because the charged free radical cation intermediate needs to be stabilized. The reaction first involves electron transfer, followed by the formation of free radical cations, and finally the free radical cations are deprotonated to form the final product. This process is related to the proton dissociation energy, and the ionization potential plays a key role in the first step of electron transfer. The lower ionization potential is conducive to the removal of electrons, thereby promoting the electron transfer process. The ionization potential usually decreases with the decrease in solvent polarity, and the introduction of hydroxyl or methoxy groups in antioxidant molecules can further reduce the ionization potential, thereby enhancing their SET ability. In the SPLET mechanism, protons are first extracted by the solvent to form antioxidant anions. Subsequently, electron transfer is performed to transfer electrons to DPPH radicals. The first step of this process depends on the proton affinity of the antioxidant anion, and the second step depends on the electron transfer energy. The whole process is relatively slow because the proton loss process is significantly affected by the pH value, so it is closely related to the pH value of the measured solution. The SET-PT mechanism is similar to the SPLET mechanism, but the difference is that the mechanism first undergoes electron transfer, followed by PT. This process is described by the ionization potential and proton dissociation energy of the free radical cation formed in the first step. Due to insufficient overlap, electron transfer is difficult to occur alone in the determination, and the nuclear and electronic structures of the transition state are not ideal. Therefore, HAT is not the only mechanism. The SET-PT mechanism is more thermodynamically favorable in polar media because polar solvents reduce the ionization potential or proton affinity of the antioxidant but have little effect on the bond dissociation energy.

## 4. Prediction of Antioxidant Functional Properties by Machine Learning

In recent years, ML, as a cutting-edge technology with significant interdisciplinary capabilities, has deeply penetrated into many fields such as food research and development, industrial manufacturing, and medical health [60]. In the traditional research, the relationships between the esterification reaction efficiency of VC and FA, the antioxidant properties of the product, and the molecular structure and synthesis processes mainly depend on many orthogonal experiments, which have the limitations of long research periods, incomplete variable coverage, and difficult quantification of multi-factor coupling rules. By introducing the experimental data into the ML model, the discrete experimental samples can be transformed into a quantitative model with predictive ability, which can accurately capture the nonlinear coupling relationships between the above factors and effectively break through the technical bottleneck of traditional research. Different from traditional programming, ML can accurately and effectively explore the nonlinear relationships between complex variables by constructing data-driven computational models, demonstrating advantages in improving prediction accuracy and intelligent decision-making [61]. In the system of ML methods, supervised learning achieves predictive modeling by constructing the mapping relationship between input features and target labels [54]. Aiming at the scarcity of labeled data in supervised learning, it has been proved that the generalization ability of the model can be significantly improved by using a semi-supervised learning framework to integrate labeled and unlabeled samples [62,63]. The current mainstream supervised learning algorithms mainly include the following: linear regression, logistic regression, decision tree, support vector machine, K-nearest neighbor algorithm, neural network, and random forest. Table 1 introduces the advantages, disadvantages, and application scenarios of the current mainstream algorithms. Among them, the random forest algorithm [64] constructs multiple heterogeneous decision tree models through the integrated learning mechanism and uses bootstrap resampling and random feature selection strategies to effectively mitigate overfitting and improve the robustness of the model. The parallel training architecture adopted by the algorithm makes it show superior computational efficiency when dealing with high-dimensional heterogeneous data [65,66].

### 4.1. Model and Algorithm

The random forest algorithm is a powerful supervised learning method that is suitable for processing high-dimensional data that does not need to reduce the dimension of the data. It uses the “bootstrap” [74] or “bagging” [75] method, and multiple decision trees are constructed by randomly selecting feature subsets [76]. In order to form an ML model, this mechanism improves the generalization ability of the model. In addition, even if there are many missing values in the data, the model can still maintain high accuracy. This feature makes it perform well when dealing with complex and incomplete data sets [77]. Because random forest can combine different independent predictors, this algorithm is not easy to overfit and has significant anti-interference ability, which can effectively resist the interference of outliers and noise [78]. At the same time, it does not have strict assumptions about the distribution of data, which makes it perform well in dealing with various types of data, and the data does not have to conform to a specific distribution model. Wenbo Chen et al. [79] conducted experiments that show that the random forest model has an excellent fitting effect, and compared with other commonly used algorithms, it has better accuracy and robustness. In addition, the embedded feature selection method of the random forest algorithm can effectively reduce its time cost.

JunHao Chen et al. [80] showed in their research that the random forest model is constructed as follows. Firstly, the bootstrapping method is used to generate a subdataset with the same size as the original dataset from the original dataset, which is used to train a single decision tree. At each split node, a feature subset is randomly selected from all the features, and the importance of these features is evaluated according to the degree of impurity reduction (such as information gain, Gini index, etc.) to determine the optimal split point, and then it iterates and repeats the process until the stop condition is satisfied to generate a decision tree. The operation process is shown in Figure 5. Finally, the above steps are repeated to train multiple decision trees, and the prediction results of multiple trees are integrated by majority voting (classification problem) or averaging (regression problem). By calculating the prediction error of each decision tree for Out-of-Bag data (OOB) samples, the performance index of each decision tree can be obtained. According to its training performance, the optimal number of learning cycles and learning rate are determined. The test data set is introduced into the trained model to evaluate its prediction accuracy [81]. The OOB errors of all decision trees in the random forest model are summarized, and the mean square error is calculated to measure the overall performance of the whole random forest model [82].

In order to optimize the prediction performance of the model, the key parameters of the model construction are adjusted. The parameters include node size (controlling the minimum number of leaf nodes in the decision tree), mtry (the number of features randomly selected for each split), and ntree (the total number of decision trees in the forest) [83]. The calculation equation is as follows:

The calculation formula of predicted value: $\hat{\;y}\left(x_{i}\right)=\frac{1}{K}\sum_{k=1}^{K}T_{D\left(\theta_{k}\right)}\left(x_{i}\right)$
$x_{i}$ represents the proportion value of the ith sample$D\left(\theta_{k}\right)$ represents the kth bootstrap sampleK is the number of each tree $\left(k=1,2,\ldots ,K\right)$.


After the model calculation is completed, in order to evaluate the effect of the data-driven strategy, the predicted values should be compared with the actual data obtained [84]. The accuracy and performance of the model can be evaluated by calculating the regression coefficient (R^2^) and error indicators, including root mean square error (*RMSE*) and mean bias error (*MBE*, or bias), through the following formulas [83].$$R^{2}=1-\frac{\sum_{i=1}^{n}{\left(Yest-Yobs\right)}^{2}}{\sum_{i=1}^{n}{\left(Yobs-\bar{Y}\right)}^{2}}$$$$RMSE=\sqrt{\frac{\sum_{i=1}^{n}{\left(Yest-Yobs\right)}^{2}}{n}}$$$$MBE=\frac{1}{n}\sum_{i=1}^{n}\left(Yest-Yobs\right)$$
where (*Yobs*) and (*Yest*) represent observations and estimates, respectively.

### 4.2. Prediction of Antioxidant Functional Properties of Substances

In the use of ML to predict the antioxidant function of esterification products, the feature importance of random forest is calculated according to the Gini coefficient [85] and characteristic parameters that can express the strength of antioxidant properties [86]. Based on the above multi-dimensional analysis, the ML model is used to predict the antioxidant properties of VC and FA after esterification. The DPPH free radical scavenging rate and esterification yield were selected as the predicted characteristic parameters in the random forest model. After determining the characteristic parameters, it is necessary to select the relevant and suitable target variables according to the nature and meaning of these characteristic parameters to ensure that the model can effectively learn the relationship between features and targets and achieve accurate prediction or classification [87]. The target variable is closely related to the feature expression. The target variable will drive the feature expression, and the feature expression will directly affect the prediction effect of the target variable. The Gini coefficient can measure the situation of the target variables in the data set and select the best features to divide the data set and improve the prediction performance of the model [88].

IC_50_ values can be used to represent the free radical scavenging rate in the DPPH free radical scavenging test [89]; that is, the DPPH free radical scavenging rate can reach 50% when the sample concentration is at this value. The lower the IC_50_ values, the higher the antioxidant performance, and the use of IC_50_ to calculate the total antioxidant activity is more accurate [90]. Due to the wide range of IC_50_ values, direct introduction into the model may lead to deviation in the model data set. Therefore, the original data should be preprocessed to compress the numerical range and make it closer to the normal distribution. In the study of antioxidant properties, the core correlation between IC_50_ values and the machine learning model is mainly reflected in the prediction task with IC_50_ values as the target variable. The machine learning model realizes the accurate prediction of the activity index by mining the complex correlation between input characteristics such as formula parameters, process conditions, component characteristics, and antioxidant IC_50_.

The esterification yield reflects the degree of esterification of active groups in the molecule. The degree of esterification will change the proportion of polar groups in the molecule, which may increase the fat solubility of the molecule, resulting in the change in hydrophobicity of the esterification product. At the same time, the introduction of the ester group (-COO-) will change the distribution of its electrons and affect the chemical properties [91], changing the strength of its antioxidant properties. Due to the obvious threshold relationship between esterification yield and the oxidation resistance, the data can be preprocessed by separation discretization in the model so that the continuous variables can be transformed into discrete variables [92]. These discrete variables can be divided into three grades: low, medium, and high in esterification yield, and their classification characteristics can be generated to reduce data complexity and facilitate subsequent data processing and analysis. At the same time, in order to reduce the deviation of data and improve the accuracy of model prediction, feature interaction models are usually used [93]. The characteristic interaction model indirectly reflects the molecular structure of the ester product and the retention degree of active groups by capturing its coupling relationships with the substrate structure and reaction process and then accurately predicts the antioxidant performance. In the esterification reaction, the number of hydroxyl (-OH) may affect the polarity, solubility, and hydrophobicity of the molecule, thus affecting the yield of the esterification reaction. The relative position of hydroxyl (-OH) and carboxyl (-COOH) in the molecular structure also has a significant effect on the esterification yield [94]. When the hydroxyl and carboxyl groups are located at the end of the molecule, the esterification yield is higher, while when they are located in the middle position, the yield may be lower. Therefore, in the feature interaction model, the number of hydroxyl groups and the relative position of hydroxyl and carboxyl groups in the compound can be selected as other independent variables. Thus, the expression ability of the model is significantly improved, and the nonlinear relationship of the data is better fitted. At the same time, this feature method not only improves the prediction accuracy and generalization ability of the model but also enhances the interpretability of the model, improves the computational efficiency, and reduces the overfitting probability [95].

Because untreated data has three core problems—feature weight bias, slow gradient descent convergence, and numerical calculation deviation—the data should be normalized before the model training is completed. Normalization is a key data preprocessing step in ML and data analysis [96]; it ensures that each feature has the same weight in the model training process by scaling the data of different features into a unified range [97]. This processing method is very important to improve the stability and efficiency of model training because it can effectively avoid the leading role of some features in model training due to dimensional differences or too large a numerical range and improve the accuracy of the model. After normalization, the data is input into the random forest model for training. In the process of model training, the performance of the model is comprehensively evaluated by calculating regression coefficients and error indices, and this evaluation reflects the fitting degree of the model to the training data. The trained model can be used to predict the antioxidant properties of VC fatty acid esters formed by esterification of VC with different liquid-phase FAs.

Therefore, ML brings multi-dimensional core added value to the esterification process. It breaks through the bottlenecks of traditional research and production by constructing a data-driven model and transforms discrete experimental data into quantitative predictive relationships, which can quickly capture the nonlinear correlations between reaction conditions, esterification efficiency, and product performance, significantly shortening the process development cycle and reducing costs. At the same time, it can accurately predict the IC50 value, ester solubility, and other key properties of ester products to guide directional synthesis. It can also establish a monitoring model by collecting reaction data in real time to improve the stability of the esterification reaction [98]. In practical applications, ML has been widely used in the optimization of functional ester synthesis processes, the screening of antioxidant properties of products, the intelligent control and quality monitoring of industrial esterification processes, and the design of new ester molecules, providing efficient support for the entire chain from laboratory research and development to industrial-scale production.

## 5. Conclusions and Future Prospects

In this paper, the esterification reaction mechanism, product characteristics, and antioxidant mechanism of VC with PA, LA, and OA, as well as the application value of ML, were systematically reviewed. It was pointed out that the esterification reaction of VC with different FAs mainly occurs at the C-6 hydroxyl group, and the active hydroxyl groups at the C-2 and C-3 positions are retained, enabling the esterified products to possess both antioxidant activity and fat solubility. Additionally, enzyme-catalyzed synthesis and GSL-MPS are more in line with green and sustainable requirements. Ascorbyl fatty acid esters can neutralize free radicals through hydrogen, proton, and electron transfer of their enediol structure. The DPPH and ABTS methods, combined with IC50 values, can comprehensively evaluate their antioxidant properties and alleviate food quality deterioration caused by oxidative stress. Meanwhile, ML, especially the random forest algorithm, was discussed. By constructing a data-driven model, using the DPPH free radical scavenging rate and esterification yield as characteristic parameters, and combining a feature interaction model with data preprocessing, high-precision performance prediction is achieved, which provides an efficient tool for the screening and design of new antioxidants. But the existing machine learning models rely on reported datasets, exhibit low prediction accuracy and weak generalization for new fatty acid-derived ascorbates and composite systems, and lack guidance for the esterification process optimization mechanism. At the same time, data set samples are unbalanced, indicators are inconsistent, and data support is insufficient, which limits the quality of model training and its application scope.

In view of the above problems, combined with the intelligent development needs of the food industry, future research needs to promote the breakthrough of ascorbate research from basic exploration to industrial application. Firstly, high-quality and wide-coverage datasets should be constructed, sample dimensions should be expanded, and performance evaluation standards should be standardized, and a standardized database should be established to provide a solid foundation for model optimization. Secondly, the model structure should be optimized, and interpretability should be improved to improve the prediction ability for complex systems. In addition, ML models should be combined with the actual needs of the food industry, and green processes such as enzyme-catalyzed esterification should be integrated to improve the preparation efficiency and reduce environmental costs. In summary, the development potential of ascorbyl fatty acid esters is huge. The in-depth application of ML technology will provide strong support for the functional and intelligent upgrading of the food industry through the integration of data-driven approaches and chemical synthesis and ultimately achieve the core goals of improving food quality and ensuring food safety.

## Figures and Tables

**Figure 1 foods-14-04255-f001:**
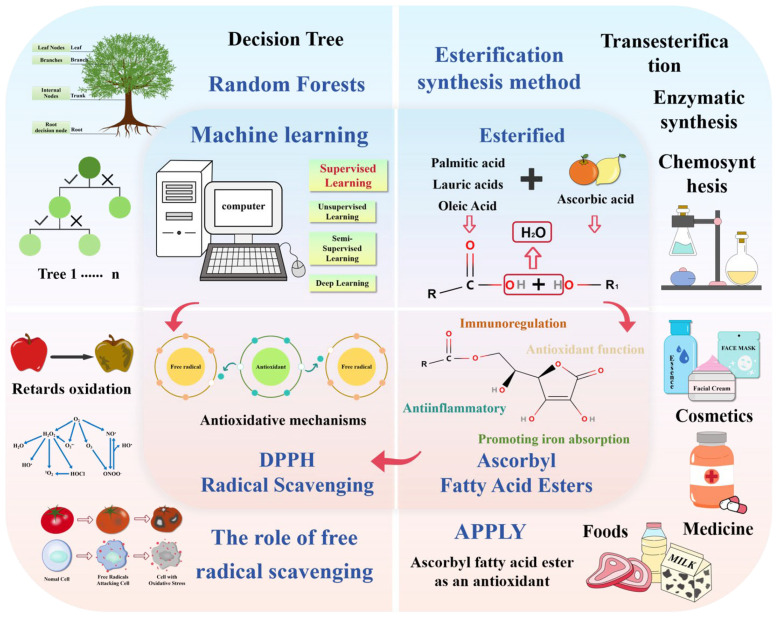
The four core contents of the article: the relationship between ML, esterification reaction, different ascorbic acid fatty acid esters, and antioxidant mechanism.

**Figure 2 foods-14-04255-f002:**
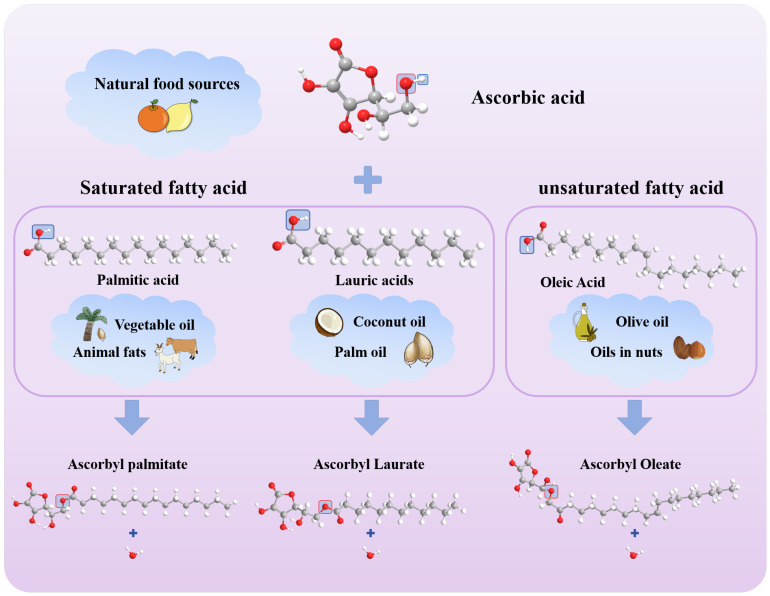
Esterification of VC with different FA (palmitic acid, lauric acid, and oleic acid). (Red represents O, gray represents C, and white represents H. After the reaction, H and OH in the blue box synthesize H2O, and O in the red box is retained to bind to fatty acids).

**Figure 3 foods-14-04255-f003:**
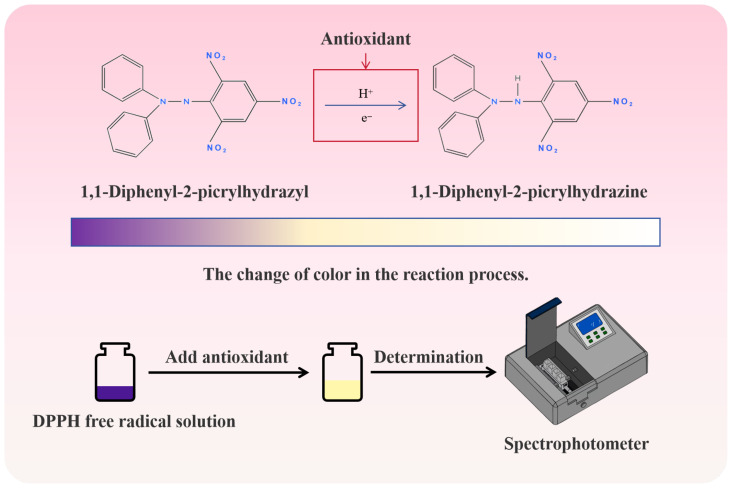
DPPH free radical scavenging principle and its color change.

**Figure 4 foods-14-04255-f004:**
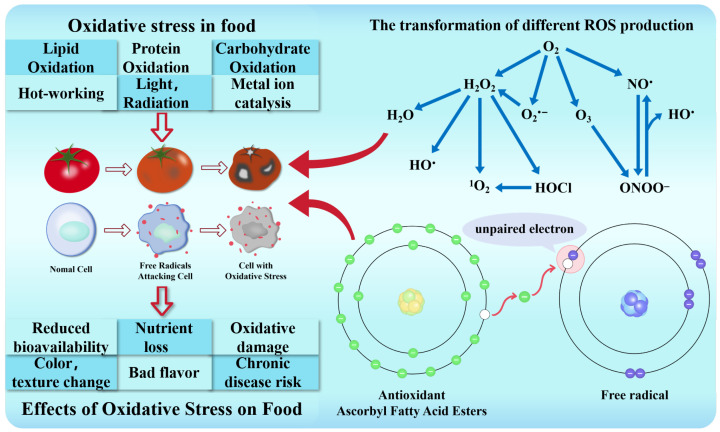
The mechanism of different ROS leading to food oxidative stress and antioxidant (VC fatty acid ester) scavenging free radicals. (Tomatoes represent the food decay process corresponding to the cell oxidation process below. Green is the electron in the antioxidant. By transferring the electron, it is paired with the unpaired electrons in the free radical to become a stable atom, thus playing an antioxidant role).

**Figure 5 foods-14-04255-f005:**
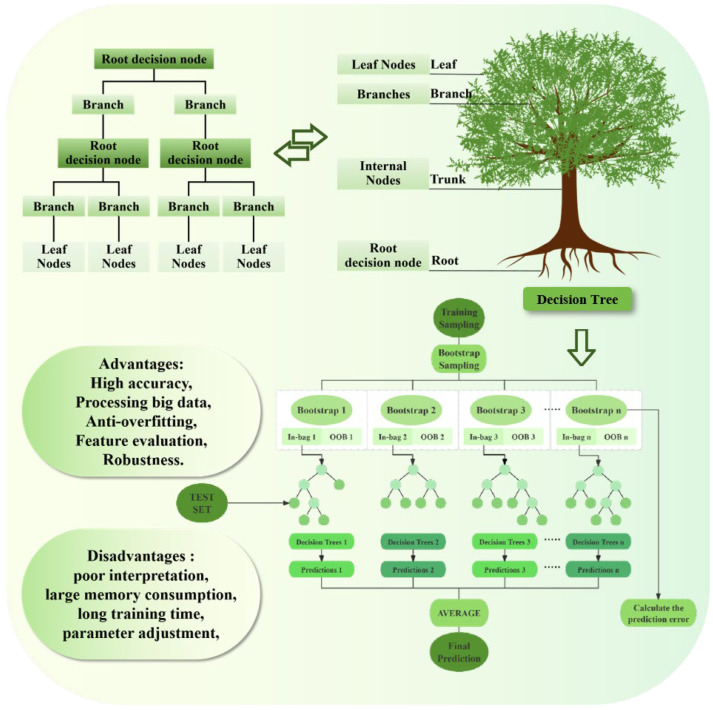
Random forest model operation process.

**Table 1 foods-14-04255-t001:** The Aominance, inferiority, and application scenarios of common supervised learning algorithms.

Algorithm	Aominance	Inferiority	Application Scenarios	Reference
Linear regression	Simple to understand, easy to implement, computationally efficient and scalable	Sensitivity to outliers, linear hypothesis, feature independence hypothesis	It is suitable for regression problems with obvious linear relationship between target variables and features.	[67]
Logistic regression	Simple and easy to understand, easy to implement, computationally efficient, output probability	Sensitive to outliers, linear hypothesis, feature independence hypothesis, poor interpretability	It is suitable for the binary classification problem with a linear relationship between features and target categories.	[68]
Decision tree	Simple and easy to understand, the model has strong interpretability, does not need feature scaling, can handle classification and regression problems, and can handle missing values	Easy to overfit, unstable and complicated to calculate	It is suitable for classification and regression problems with fewer features and obvious decision rules between features.	[69]
Support vector machine	Strong generalization ability, can handle high-dimensional data, and good robustness	Computationally complex, sensitive to parameter selection, susceptible to noise, and not suitable for large-scale data	It is suitable for the classification problem with high feature dimension and obvious interval between categories.	[70]
K-nearest neighbor algorithm	Easy to understand, adaptable, no training process	High computational cost, is sensitive to data distribution, and is susceptible to noise	It is suitable for classification and regression problems with small amount of data and low feature dimension.	[71]
Neural network	Strong expression ability, strong adaptability and parallel computing	The training is complex, easy to overfit, and the model interpretation is poor	It is suitable for complex classification and regression problems with large amount of data and high feature dimension, such as image recognition, natural language processing and so on.	[72]
Random forest	Strong generalization ability, can handle large-scale data, feature importance evaluation, and can handle missing values	The model is complex, the computational cost is high, and the memory consumption is large	It is suitable for classification and regression problems with more features and large amount of data, especially when there is a complex interaction between features.	[73]

## Data Availability

No new data were created or analyzed in this study. Data sharing is not applicable to this article.

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
