# Peer review of "Prediction of Esterification and Antioxidant Properties of Food-Derived Fatty Acids and Ascorbic Acid Based on Machine Learning: A Review"

_foods, 2025, doi:10.3390/foods14244255_

Round 1

Reviewer 1 Report

Comments and Suggestions for Authors

This is a great and timely study in our field. Please address the comments to further strengthen the manuscript. 
It needs edit in several sections, specifically - figures, it should be within the margins; tables, in the same font and size and within the margins; the credit sections have different fonts and sizes used.

Comments on the Quality of English Language

The sentences and use of the language need to be significantly improved. 

Author Response

Reply to the comments by Reviewer 1

Thank you very much for your comments. We carefully read your comments, and our revisions and replies are as follows.

Comment 1:The graphics in the abstract are great, but should it be included in a different section? Have not seen a graphical abstract earlier.

Respond 1:  Thank you for your valuable comments. We have inserted the summary diagram into lines 117-119 of the article 's introduction and marked it red for you to review.

Comment 2: First sentence – the structure should be improved, ‘It’ does not need to be capitalized.

Respond 2: Thank you for your valuable comments. We have modified the sentences of line 37 and marked it red.

Comment 3: Did not understand this statement: “and the content of ascorbic acid in natural foods (10-100 mg / 100 g) is usually higher than the content of synthetic ascorbic acid”. Synthetic ascorbic acid content is less where...?

Respond 3: Thanks for your reminder. We have modified this sentence in lines 39-41 and marked it red.

Comment 4: The first paragraph, use of ‘Ascorbic acid’ is redundant, maybe use the acronym that has been defined as VC?

Respond 4: Thanks for your reminder. We have modified the ' ascorbic acid ' that appears after the full text is defined as ' VC ' to an acronym and marked it red.

Comment 5: Ascorbic acid reduces ROS to less reactive species, where is the phenomena helpful? The sentences need to be explained more clearly.

Respond 5: Thank you for your question. We added a sentence to explain this phenomenon in lines 42-46 of the article and marked it red.

Comment 6: The second and third paragraphs describe FA; there is no clear link to why it is explained after Ascorbic acid. Please establish a clear reasoning for introducing FAs at the beginning of the paragraph.

Respond 6: Thank you for your valuable comments. The combination of fatty acids and ascorbic acid can produce a synergistic effect, and its overall antioxidant activity is significantly improved compared with the use of two substances alone. We have detailed the above in line 51-57, and marked it red.

Comment 7: In paragraph 4: “Ascorbic acid is a strong antioxidant water-soluble vitamin”. This has already been explained at the beginning.

Respond 7: Thanks for your reminder. We modified this sentence in lines 83-89 as related to the synergistic effect of ascorbic acid and fatty acids, and marked it red.

Comment 8: Figures should be placed within the margins! This needs to be edited.

Respond 8: Thanks for your reminder. We modified lines 117, 173, 417, 436, 569, the position of the graph, consistent with the rest of the article, and marked it red.

Comment 9: Palmitic acid – “which is very rich in nature.” This can be improved in structuring.

Respond 9: Thanks for your reminder. We modified the structure of this sentence in line 177 and marked it red.

Comment 10: Cite for this section: “By maintaining the stability and fluidity of cell membranes, palmitic acid is essential for normal cellular physiological functions. Its antioxidant components effectively scavenge free radicals and reduce oxidative damage. In addition, palmitic acid can also promote the absorption of fat-soluble vitamins to improve the utilization efficiency of fat-soluble vitamins in the body.”

Respond 10: Thank you for your valuable comments. In line 183-186, we quoted the relevant literature to support this view, and on this basis, we made a moderate adjustment to the original sentence, enhanced its logical coherence, and marked it red.

Comment 11: AP paragraph – it is incorrect to use capital letters “Soluble in ethanol” in between a sentence. Also explain in the beginning why you are explaining about AP here.

Respond 11: Thanks for your reminder. We modified the content in line 187-188, added context coherence, and painted it red.

Comment 12: What is DPPH? Please explain.

Respond 12: Thanks for your reminder. We explained DPPH in detail in lines 388-394, and added a picture in lines 408-414 to visualize the DPPH free radical scavenging principle and mark it red.

Comment 13: Please correctly structure the sentence: “Issa Javidipour et al. (Javidipour et al., 2013), the effects of ascorbyl palmitate on per-oxide value and malondialdehyde content of refined cottonseed oil and virgin olive oil under specific conditions were studied.”

Respond 13: Thanks for your reminder. We added this sentence in lines 326 - 328 and marked it red.

Comment 14: Please explain ABTS, DMPD, etc. All acronyms should be explained before they are used. It is explained later in the manuscript. This should be done before using the acronyms.

Respond 14: Thank you for your question. We added the full name of the three abbreviations of ABTS, DPPH and DMPD in lines 375-377 and marked it red.

Comment 15: The tables are not in the same format as rest of the paper, please follow the journal requirements. Ideally keep it in the same font and size as the rest of the paper and within the margins.

Respond 15: Thanks for your reminder. We unified the format of the table with the rest of the paper in lines 515-516, and marked it red.

Comment 16: In the introduction section, acronyms for different acids have been mentioned but are not consistently used. Once it is defined, it is easier to form sentences with acronyms, read and understand too.

Respond 16: Thanks for your reminder. We have been mentioned in the article has been referred to as acronym of fatty acids, etc.all modified to abbreviations, and it marked red.

Comment 17: Some paragraphs in the introduction section do not have a clear linking why it is mentioned. It needs to be explained in the beginning the need of that section to give a holistic view to the readers.

Respond 17: Thank you for your question. We added a brief introduction to the background of ascorbic acid in lines 31-37. In lines 83-89, the content on the synergistic effect of ascorbic acid and fatty acid esterification was added to increase the relevance of the introduction. In lines 95-100, the machine learning model was supplemented to predict the antioxidant properties of ascorbyl fatty acid esters, making the context more coherent and marking it red.

Reviewer 2 Report

Comments and Suggestions for Authors

Dear Authors, you should address my comments highlighted across the text.

My comments related to the manuscript ‘Prediction of esterification and antioxidant properties of food-derived fatty acids and ascorbic acid based on machine learning: A review' are reported below:

- The present review is based on the use of machine learning, a branch of artificial intelligence, to manage the esterification process between ascorbic acid and fatty acids, which represents an important topic related to the food industry.

- Abstract: Some amendments should be made, particularly referred to a sentence highlighted whose meaning is not clearly understandable.

- Keywords: Some keywords should be replaced with new ones different than those included in the title.

- 1. Introduction: This section has been developed with sufficient details regarding the review topic, though a few modifications are needed.    

- 2. The process of esterification of ascorbic acid and fatty acid: The three sub-sections included in this section need to be revised in the first paragraphs because of the bad English phrasing, though the contents are exhaustive.

- 3. Antioxidant properties and mechanism: This section and the related sub-sections have been satisfactorily detailed but some comments should be addressed.

- 4.   Prediction of antioxidant functional properties by machine learning: The topic of this section is interesting and well developed, appropriately including a table reporting the Aominance, inferiority and application scenarios of common supervised learning algorithms.

- 5. Conclusion and future prospects: This section suggest the current framework and future perspectives of the subjects developed in this review, though it needs to be revised to better amalgamate the whole content.  

- References: The citation formatting either in this section or across the text should be checked based on this Journal style.

- The English language should be deeply revised all over the manuscript.

Comments on the Quality of English Language

The English must be improved to clearly express the concepts.

Author Response

Reply to the comments by Reviewer 2

Thank you very much for your comments. We carefully read your comments, and our revisions and replies are as follows.

Thank you for revising the article 's errors in the PDF, and for all the questions you have raised, we have revised it in the article and marked it red.

Comment 1: Abstract: Some amendments should be made, particularly referred to a sentence highlighted whose meaning is not clearly understandable.

Respond 1: Thanks for your reminder. We modified the abstract in lines 13-26, highly condensed the content of this article, and marked it red.

Comment 2: Keywords: Some keywords should be replaced with new ones different than those included in the title.

Respond 2: Thank you for your valuable comments. In line 27, we replaced the keyword that coincides with the title of this article and marked it red.

Comment 3: Introduction: This section has been developed with sufficient details regarding the review topic, though a few modifications are needed.

Respond 3: Thanks for your reminder. In the introduction of the 31-37, 51-57, 83-89, 95-100 lines added some content, enhance the relevance of the content of the article, and it marked red.

Comment 4: The process of esterification of ascorbic acid and fatty acid: The three sub-sections included in this section need to be revised in the first paragraphs because of the bad English phrasing, though the contents are exhaustive.

Respond 4: Thanks for your reminder. We modified the error on lines 177, 189, 198 and marked it red.

Comment 5: Antioxidant properties and mechanism: This section and the related sub-sections have been satisfactorily detailed but some comments should be addressed.

Respond 5: Thank you for your valuable comments. We added two free radical scavenging experiments in lines 354-371, which is more complete in this part. In lines 408-414, DPPH was explained in detail, and a diagram was added in lines 417-418 to visualize the DPPH free radical scavenging principle and mark it red.

Comment 6: Prediction of antioxidant functional properties by machine learning: The topic of this section is interesting and well developed, appropriately including a table reporting the Aominance, inferiority and application scenarios of common supervised learning algorithms.

Respond 6: Thank you very much for your recognition of this part.

Comment 7: Conclusion and future prospects: This section suggest the current framework and future perspectives of the subjects developed in this review, though it needs to be revised to better amalgamate the whole content.

Respond 7: Thanks for your reminder. In lines 647-679, we integrated the content of this article, revised the conclusions and prospects, and marked it red.

Comment 8: References: The citation formatting either in this section or across the text should be checked based on this Journal style.

Respond 8: Thanks for your reminder. We have modified the references according to the journal format.

Comment 9: The English language should be deeply revised all over the manuscript.

Respond 9: Thanks for your reminder. We have reviewed the full text and corrected errors such as grammar.

Reviewer 3 Report

Comments and Suggestions for Authors
  • In the revised paper, please add line numbering to facilitate our review. 
  • The relevance of this review is not well stated by authors in the Abstract neither in the Introduction.
  • In the Abstract, authors should clearly describe the sections of the review and draw the main conclusion.
  • Overall, the Abstract must be revised and rewritten.
  • The graphical abstract is not clear and not evident. Please change it with another one more clear and smooth.
  • Why "ascorbic acid" and "fatty acid"? Which common points and it's only one fatty acid or all fatty acids? (section 2).
  • The paper is not well illustrated, especially for a review: only 2 tables and 2 figures. 
  • The English of this paper should be significantly improved. 
  • The interaction between fatty acids and Ascorbic acid is not well detailed in the Introduction, leading to a not strengthen justification of this review! please elaborate more this synergy and what is the knowledge that this review aim to raise?!
  • The transition to ML paragraph in the Introduction is not coherent and should be revised. 
  • Also, authors did not link what is the added value of ML to this review and what is the state of art in terms of using ML in such field of chemistry (esterification). 
  • Please introduce Figures before its presentation: e.g. Section 2 starts with a figure and this is not appreciated in scientific papers. 
  • Also, elaborate a more comprehensive paragraphs to describe the importance of esterification in food science and technology. 
  • Before describing the 3 fatty acids, provide an overview on range of existing fatty acids and why did you choose only oleic, palmitic and lauric. 
  • Generally, the paper lacks of coherence between its paragraphs and sections. Authors should address this issue. 
  • Besides the subsection 4.3, The ML section is too general and not related to the core topic of this review (esterification process). In addition the subsection 4.3 is not well written (what is IC50? which relation with ML?Normalization? ...).

Author Response

Reply to the comments by Reviewer 3

Thank you very much for your comments. We carefully read your comments, and our revisions and replies are as follows.

Comment 1: In the revised paper, please add line numbering to facilitate our review.

Respond 1: Thanks for your reminder. We have added line numbers to the original text for your review

Comment 2: The relevance of this review is not well stated by authors in the Abstract neither in the Introduction.

Respond 2: Thanks for your reminder. We modified the abstract in lines 13-26, and the content of this article is highly concise. In the introduction of the 31-37, 51-57, 83-89, 95-100 lines added some content, enhance the relevance of the content of the article, and marked it red.

Comment 3: In the Abstract, authors should clearly describe the sections of the review and draw the main conclusion.Overall, the Abstract must be revised and rewritten.

Respond 3: Thank you for your valuable comments. We have modified and rewritten the content of the abstract in lines 13-26 and marked it red.

Comment 4: The graphical abstract is not clear and not evident. Please change it with another one more clear and smooth.

Respond 4: Thanks for your reminder. We inserted a clearer figure again in lines 117, 173, 417, 436 and 569, and marked it red.

Comment 5: Why "ascorbic acid" and "fatty acid"? Which common points and it's only one fatty acid or all fatty acids? (section 2).

Respond 5: Thank you for your question.

(1) Ascorbic acid and fatty acids are natural substances, the combination of the two can produce antioxidant synergy, while improving the chemical properties of a single substance. We have added the above content in line 83-89 and marked it red.

(2) We supplemented the types of fatty acids in detail in lines 121-147, and the representativeness and importance of these three fatty acids in food science, nutrition and health, and industrial applications, and marked them red.

Comment 6: The paper is not well illustrated, especially for a review: only 2 tables and 2 figures.

Respond 6: Thank you for your valuable comments. We added a picture in lines 417-418 to explain the experimental principle of DPPH free radical scavenging and marked it red.

Comment 7: The English of this paper should be significantly improved.

Respond 7: Thanks for your reminder. We have reviewed the full text and corrected errors such as grammar.

Comment 8: The interaction between fatty acids and Ascorbic acid is not well detailed in the Introduction, leading to a not strengthen justification of this review! please elaborate more this synergy and what is the knowledge that this review aim to raise?!

Respond 8: Thank you for your valuable comments. Esterification of ascorbic acid and fatty acids can improve the chemical properties of ascorbic acid that is easily oxidized. Compared with the use of two substances alone, its overall antioxidant activity is significantly improved. We elaborate on the above content in 51-57 lines and mark it red.

Comment 9: The transition to ML paragraph in the Introduction is not coherent and should be revised.

Respond 9: Thanks for your reminder. We supplemented the machine learning model to predict the antioxidant properties of ascorbyl fatty acid esters in lines 95-100, making the context more coherent and marking it red.

Comment 10: Also, authors did not link what is the added value of ML to this review and what is the state of art in terms of using ML in such field of chemistry (esterification).

Respond 10: Thanks for your reminder. The traditional orthogonal experiment explores the structure, process and antioxidant activity of esterification, which has a long cycle, narrow dimension and difficult to quantify. The machine learning model can be used to accurately predict and break through the technical bottleneck of traditional research. We made a detailed supplement to the above content in 486-495 lines and marked it red.

Comment 11: Please introduce Figures before its presentation: e.g. Section 2 starts with a figure and this is not appreciated in scientific papers.

Respond 11: Thanks for your reminder. We moved Figure 2 to 173-174 lines, and added the relevant content about fatty acids and esterification in front of it, making the chapter more coherent, the content more abundant, and marking it red.

Comment 12: Also, elaborate a more comprehensive paragraphs to describe the importance of esterification in food science and technology.

Respond 12: Thank you for your question. Esterification technology runs through food scenes such as flavor, wine, oil and preservation. Enzyme-catalyzed green esterification is replacing high-pollution chemical methods, providing core support for safe and efficient production of functional esters and industrial low-carbon transformation. We supplemented the above content in detail in lines 149-171 to illustrate the importance of esterification in food science and marked it red.

Comment 13: Before describing the 3 fatty acids, provide an overview on range of existing fatty acids and why did you choose only oleic, palmitic and lauric.

Respond 13: Thank you for your question. We detailed in lines 121-148 added, based on the classification of carbon chain length fatty acids, and select the three fatty acids were class representative of the long chain, medium chain, short chain fatty acids, and marked it red.

Comment 14: Generally, the paper lacks of coherence between its paragraphs and sections. Authors should address this issue.

Respond 14: Thank you for your valuable comments. We added content to lines 31-37, 51-57, 83-89, 95-100, 187-188, 240-245, 290-292, and 354-371, increasing the logicality of the context and strengthening the relevance of the full text, and marked it red.

Comment 15: Besides the subsection 4.3, The ML section is too general and not related to the core topic of this review (esterification process). In addition the subsection 4.3 is not well written (what is IC50? which relation with ML?Normalization? ...).

Respond 15: Thanks for your reminder. We added related content in 595-600 lines and 631-633 lines, explained IC50 and the relationship between IC50 and machine learning in detail, and marked it red.

Round 2

Reviewer 1 Report

Comments and Suggestions for Authors

Thank you for addressing the comments.

Comments on the Quality of English Language

The sentences and use of the language need to be significantly improved. 

Author Response

Comment 1:The sentences and use of the language need to be significantly improved.

Respond: Thank you for your reminder. We have revised the grammar and other issues of the full text and marked them red.

Reviewer 2 Report

Comments and Suggestions for Authors

Dear Authors, the manuscript can be accepted for publication in Foods, in my opinion.

Author Response

Thank you very much for your appreciation. In order to further optimize our article, we revised it and highlighted it in red for better review. Sincerely thanks again.

Reviewer 3 Report

Comments and Suggestions for Authors

The paper was significantly improved. However:

  • The review still did not elucidate in a clear manner the added value AI and ML in esterification process and its reel applications... Please address this comment!
  • The review develop generalities about ML and AI (e.g. L615-L626). Please keep in the revised manuscript only details related to the core objective of this review. 
  • Add in the "abstract" some insights about the application of esterification process in the industry to well justify your investigation. 
  • L13: systematically is not suitable.
  • The "Conclusion" section should be summarized. 

Author Response

Thank you very much for your comments. We carefully read your comments, and our revisions and replies are as follows.

Comment 1:The review still did not elucidate in a clear manner the added value AI and ML in esterification process and its reel applications... Please address this comment!

Respond: Thank you for your valuable comments. We 've added to lines 654-668 about the value of machine learning in industry and its applications, and marked it red.

Comment 2: The review develop generalities about ML and AI (e.g. L615-L626). Please keep in the revised manuscript only details related to the core objective of this review.

Respond: Thank you for your valuable comments. We have retained the core content related to this review, and supplemented the content in lines 621-625 and marked it red.

Comment 3: Add in the "abstract" some insights about the application of esterification process in the industry to well justify your investigation.

Respond: Thank you for your question. We have added relevant content on the industrial application of esterification process in lines 18-20 and marked it red.

Comment 4: L13: systematically is not suitable.

Respond: Thanks for your reminder. We have modified line 13 and marked it red.

Comment 5: The "Conclusion" section should be summarized.

Respond: Thank you for your reminder. We have made changes to the conclusion section in line 670-705, summarizing the content of the article more comprehensively and marked it red.
